# ReffAKD: Resource-efficient Autoencoder-based Knowledge Distillation

**Divyang Doshi**
Department of Computer Science
North Carolina State University
ddoshi2@ncsu.edu

**Jung-Eun Kim**
Department of Computer Science
North Carolina State University
jung-eun.kim@ncsu.edu

## Abstract

In this research, we propose an innovative method to boost Knowledge Distillation efficiency without the need for resource-heavy teacher models. Knowledge Distillation trains a smaller "student" model with guidance from a larger "teacher" model, which is computationally costly. However, the main benefit comes from the soft labels provided by the teacher, helping the student grasp nuanced class similarities. In our work, we propose an efficient method for generating these soft labels, thereby eliminating the need for a large teacher model. We employ a compact autoencoder to extract essential features and calculate similarity scores between different classes. Afterward, we apply the softmax function to these similarity scores to obtain a soft probability vector. This vector serves as valuable guidance during the training of the student model. Our extensive experiments on various datasets, including CIFAR-100, Tiny Imagenet, and Fashion MNIST, demonstrate the superior resource efficiency of our approach compared to traditional knowledge distillation methods that rely on large teacher models. Importantly, our approach consistently achieves similar or even superior performance in terms of model accuracy. We also perform a comparative study with various techniques recently developed for knowledge distillation showing our approach achieves competitive performance with using significantly less resources. We also show that our approach can be easily added to any logit based knowledge distillation method. This research contributes to making knowledge distillation more accessible and cost-effective for practical applications, making it a promising avenue for improving the efficiency of model training.

## 1 Introduction

Convolution Neural Network (CNN) [13] [11] [23] [24] [6] have achieved remarkable success in various computer vision tasks, including image classification, object detection, and image segmentation. These networks have grown in complexity and depth, thanks to advancements in GPU and computational resources. Models like ResNet [6] [31] [27] and Vision transformers [4] [25] have become increasingly large and accurate. However, hosting such models on resource-constrained systems like embedded and edge platforms is a challenge.

One solution is Knowledge Distillation (KD) Fig. 2, where a large pre-trained "teacher" model guides a smaller "student" model to match its performance. However, KD has drawbacks. It requires a large, trained teacher model, which can be computationally expensive. Yet, KD's effectiveness mainly stems from "soft labels" that convey class similarity and enrich the student model's training beyond what hard labels offer.

Workshop on Advancing Neural Network Training at 37th Conference on Neural Information Processing Systems (WANT@NeurIPS 2023).

In this study, we explore an efficient, unsupervised method to identify class similarities and generate high-quality soft labels for KD. We use *autoencoders*, neural networks that create compact image representations by reconstructing data. These representations capture essential features in a single-dimensional vector. We calculate a similarity matrix using cosine similarity and feed it as soft labels to the student model during training. Our experimental results on CIFAR-100 [10], Tiny Imagenet [12], and Fashion MNIST [29] datasets demonstrate that our approach is able to achieve competitive performance to those of recent and traditional KD techniques while requiring significantly less resources. In summary, our contributions are as follows:

- We developed a novel approach to knowledge distillation without the need to use a large teacher model while maintaining competitive (or better) performance.
- We compared the resource footprints, FLOPs (FLoating point Operations), MACs (Multiply–Accumulate operations)[6], Memory [20], and parameter counts [23], showing the efficiency of our approach.
- We showed how our method works with existing advancements in KD, and how it's better/comparable to some of the existing methods of both logit-based and feature-based KD.

## 2  Related Work

There has been a lot of research on utilizing soft labels to improve the performance of smaller networks. Wang et al. [28] analyzed the effect of using a very large model for distillation. They show that using an extremely large model for distillation can result in a smaller model that does not generalize well because of high confidence in the soft labels of the teacher model. They also proposed a novel pruning approach to help improve the generalization performance when using an extremely large model for distillation. Collins et al.[2] showed that using soft labels directly to train a model can achieve comparable performance to other KD techniques. They also developed and released a new dataset CIFAR-10S which consists of soft labels from a crowdsourcing approach. Peterson et al.[19] and Uma et al.[26] also discuss using crowd-sourced soft-labeled data to improve the performance of deep neural networks in several machine learning tasks.

Recent work includes "Decoupled Knowledge Distillation" (DKD) by Zhao et al. [32], which separates knowledge transfer into target and non-target class distillation while emphasizing logit distillation's importance. Some older work by Passalis et al. [18] introduced a novel probabilistic knowledge transfer method, outperforming existing techniques and expanding knowledge transfer applications. Additionally, Lee et al. [14] introduced a method that enhances knowledge distillation with singular value decomposition, achieving better classification accuracy at reduced computational costs through self-supervised learning. Another interesting research direction is presented in the work by Yuan et al. [30] that demonstrates that the power of KD is not only because of class similarity given by the teacher model but also due to the regularization effect produced by soft labels. They focus on creating a teacher-free KD via manually designed regularization distribution from the hard labels.

Another study by Beyer et al. [1] states that vanilla distillation still works great in practice and is very effective in compressing the model size. Their findings state that using the teacher model with the same data augmentation and transformation to generate soft probabilities makes the KD work much better. The reason researchers avoid doing this is to save compute space and training time while training the smaller student model, resulting in the efficiency of KD. They also found that KD requires a large number of training epochs to converge to the best accuracy. Our approach creates a single similarity matrix that contains soft probabilities for each class. Hence, it removes the need to generate output soft probabilities per the transformation applied to each input image. Therefore, it saves a lot of computing space and time while training the student model and giving the required amount of supervision.

Another class of KD approach uses a hidden representation of the teacher models in addition to output logits given by the teacher. This class of methods was first introduced by Romero et al. in the paper FitNets [21]. This technique was further researched in work by Yifan et al. [15], Rosenfeld et al. [22], Zagoruyko et al. [9], and Park et al. [17]. It provides hints to the student model about the hidden representation of the teacher model. A hint is defined by the output of the teacher's network for a hidden layer. This type of KD is implemented through the use of a loss term, which guarantees that the student's hidden representation at different levels within the network is aligned with the teacher's

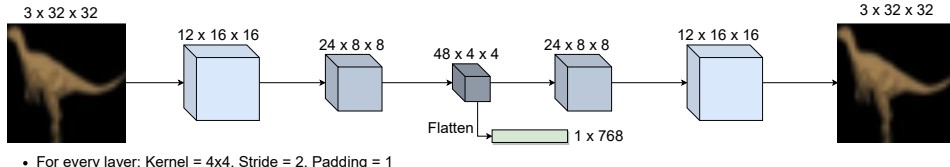

- For every layer: Kernel = 4x4, Stride = 2, Padding = 1

Figure 1: Convolutional Autoencoder for CIFAR-100.

hints. The major complexity of the approach lies in choosing the best layers in the teacher to give the hints and choosing the layers in the student model that will receive the hints.

We primarily focus on improving the logit-based knowledge distillation [7] [5] [16] [32] [18] [14] and removing the need of cumbersome teacher model to understand the underlying class similarities in the data. We also aim to improve the performance of the student models using minimal resource overhead. To our best knowledge, no other study has used autoencoders to generate soft labels based on inter-class similarity to perform knowledge distillation. Our work primarily focuses on the development of a novel method to generate soft labels without the need for a bigger network or crowd-sourced labeling by utilizing the information from the training dataset.

## 3 Methodology

### 3.1 Architecture

In our work, we developed an efficient Convolutional Autoencoder (CAE) to produce image feature representations for calculating soft probability distributions. The encoder consists of three convolutional layers with 4x4 kernels and 12, 24, and 48 filters, each with a padding of 1 and a stride of 2. The bottleneck layer provides a 768-dimension feature vector for CIFAR-100, a 3072-dimension vector for Tiny Imagenet, and a 48-dimension vector for Fashion MNIST. The decoder mirrors the encoder architecture.

We carefully examined various model configurations as part of our effort to optimize the autoencoder architecture for soft probability generation and performance improvement. As a starting point, we used a three-layer encoder and decoder design. We incrementally added convolutional layers to both the encoder and decoder in order to investigate the possible benefits of larger models. The model's training stability was negatively impacted by the difficult vanishing gradient problem created by this augmentation. Hence, we added skip connections—inspired by residual connections [6]—to facilitate gradient flow in order to address this problem. Then, in order to balance model complexity with effectiveness, we evaluated the trade-offs. Our tests showed that expanding the model past the three-layer encoder-decoder design did not result in satisfactory performance gains. With the simple three-layer autoencoder, we consistently saw good soft probability distributions and overall superior performance. As a result, we used this simplified design for our subsequent tests, placing a focus on the effectiveness of soft label production. This design decision supports the main goals of this work that is to improve resource utilization in case of knowledge distillation.

### 3.2 Soft Labels with Autoencoder

Our work focuses on a novel approach to efficiently generate soft labels for images through unsupervised learning using autoencoders. Our approach sets itself apart from simply generating random soft probability distributions based on hard labels [30]. The key distinction lies in the way our approach leverages autoencoders to encapsulate rich information about image features and their corresponding classes.

When training an autoencoder, it learns to encode the input images into a compact hidden representation. What makes this representation unique is its ability to implicitly capture characteristics that distinguish between different classes. In other words, the autoencoder's hidden layer is sensitive to the underlying features that define each class.

This inherent class-awareness of the hidden representation serves as the foundation for our soft label generation process. By utilizing these learned vectors to create soft probability distributions, we

effectively mimic the behavior of a teacher model. In essence, our approach replicates the role of a knowledgeable teacher conveying, to the best of its own understanding, the intricate relationships between different classes to the student model.

Hence, it becomes clear that our approach falls within the realm of knowledge distillation rather than being a generic soft label generator or label smoothing regularizer. We harness the power of autoencoders to distill valuable knowledge about class relationships, ensuring that our soft labels are not only probabilistic but also informed by meaningful class distinctions. This approach not only enhances the efficiency of knowledge transfer on resource-scarce platforms but also preserves performance.

To train CAE, we use stochastic gradient descent (SGD) with a momentum of 0.9, weight decay of $5e - 4$, and mean squared error as a loss function. We used an initial learning rate of 0.1 with step learning rate decay of 0.2 after 60, 120, and 160 epochs. After 200 epochs of training, we randomly selected 40 samples from each of the different classes in the dataset, resulting in a total of $40 \times c$ samples where $c$ is the number of classes.

For the next step, we construct a $c \times c$ matrix to represent the similarity score between each class in the dataset. Each row of the matrix corresponds to a specific class, and each column represents the similarity score between that class and the other classes. We compute the cosine similarity between the embedding vectors of the $40 \times c$ randomly selected samples and assign the resulting value to the corresponding element in the matrix. For instance, suppose we calculate a cosine similarity between sample 1 in class 1 and sample 2 in class 2. We add the resulting value to the cell belonging to row 1 and column 2 of the matrix, as well as to the cell belonging to row 2 and column 1 of the matrix. By repeating this process for all possible combinations of the samples, the entire matrix is built with similarity scores between every class in the dataset. Finally, we take the average of all cells in the matrix and apply softmax on each row to represent the similarity score for that cell.

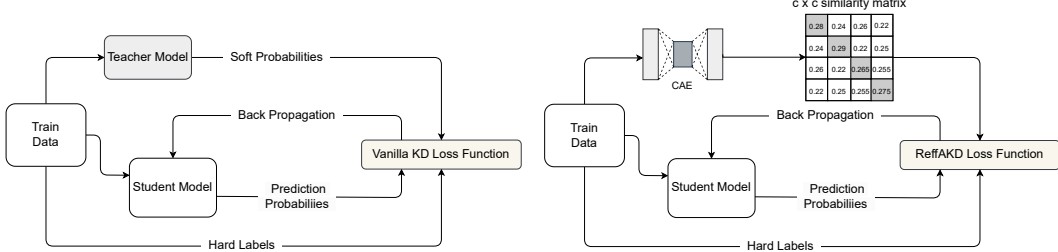

Figure 2: Vanilla knowledge distillation          Figure 3: ReffAKD distillation

To further refine the matrix, we introduce a hyper-parameter, $\gamma$, which we add to the diagonal cells of the matrix to boost the similarity score for the same class. This ensures that samples belonging to the same class have a higher similarity score than samples belonging to different classes, which is an essential criterion for generating effective soft labels. Each cell of the similarity matrix is calculated as follows:

$$z_{ij} = \begin{cases} \frac{\sum_{\forall x \epsilon i, \forall y \epsilon j} \text{cosine\_similarity}(x,y)}{\sum_{\forall x \epsilon i, \forall y \epsilon j} 1}, & \text{if } i \neq j \\ \frac{\sum_{\forall x \epsilon i, \forall y \epsilon j} \text{cosine\_similarity}(x,y) + \gamma}{\sum_{\forall x \epsilon i, \forall y \epsilon j} 1}, & \text{if } i = j \end{cases} \tag{1}$$

where $z_{ij}$ refers to the average similarity scores between class i and class j. We use grid search to find the minimum value of $\gamma$ that makes the soft label 100% accurate, i.e. probability or similarity score for $z_{ii}$ for $i \epsilon [0, c-1]$ is the highest for that row. Fig. 3 shows the complete flow of `ReffAKD`

### 3.3 ReffAKD Loss Function

Performing KD using a high-temperature value on the teacher model logits has shown to be an effective way to make the student learn better. The reasoning behind this is that softer probability distribution results in smoother gradients which in turn prevent the student from getting stuck in local minima. Furthermore, the softer probability distribution generated by the teacher model can help prevent overfitting of the student model and help it learn the underlying patterns of the data. This can lead to better generalization performance of the student model on unseen data.

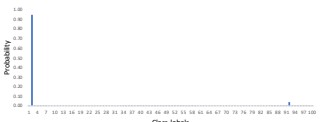

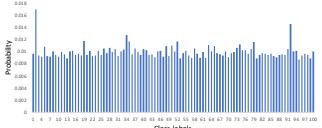

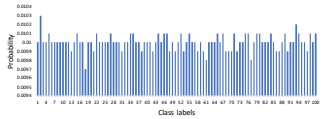

(a) ResNet50 Teacher model's output probability, temperature = 1

(b) ResNet50 Teacher model's output probability, temperature = 20

(c) `ReffAKD` output probability, temperature = 1

Figure 4: Output probability distribution comparison for ResNet50 and ReffAKD for one instance of CIFAR-100 dataset

With our approach, the soft labels generated after performing softmax over the cosine similarity matrix result in a flattened probability distribution as shown in Fig. 4. The soft probability distribution generated by our approach is comparable to the probability distribution generated by a large teacher model with a high temperature. Due to this reason, we slightly tweak the KD loss function by removing the temperature parameter that is applied to teacher model outputs (autoencoder-generated soft labels in our case.) The loss function, $L_{ReffAKD}$, is as follows:

$$
\begin{aligned}
L_{ReffAKD} = &[KLD(SO\_t, \text{AESL}) * (\alpha * t^2)] + \\
&[CE(\text{student\_output}, \text{hard\_labels}) * (1 - \alpha)]
\end{aligned}
\tag{2}
$$

where, $\text{AESL} = \sigma(\text{autoencoder\_similarity\_values})$, $SO\_t = \text{student\_output}/t$..

## 4 Evaluation

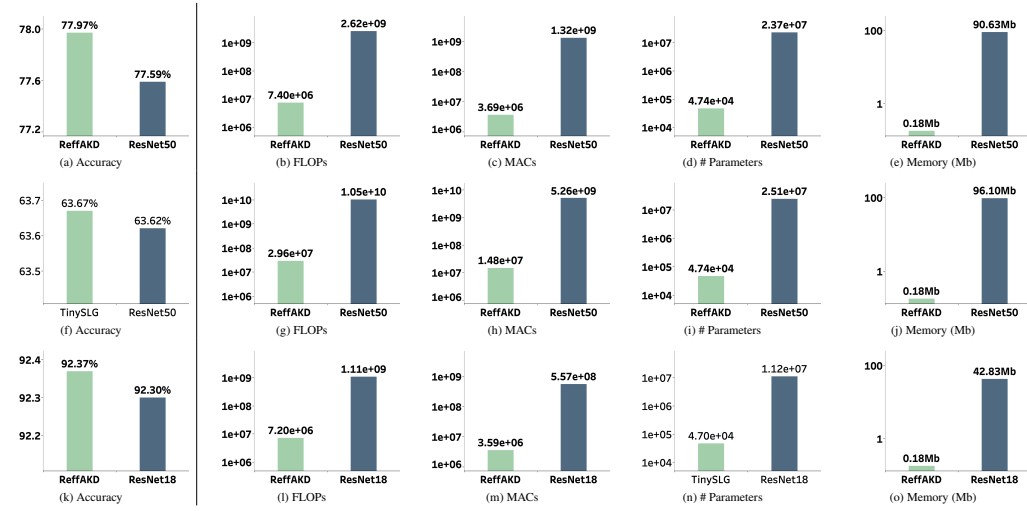

Figure 5: Accuracy and resource consumption comparisons of `ReffAKD` and vanilla KD teacher models for CIFAR-100 (first row), Tiny ImageNet (second row), and Fashion MNIST (third row). Note the Y-axes of resource consumption are in a *log* scale.

### 4.1 Experimental Setup

Our experimental results are shown with three benchmark datasets, i.e CIFAR-100, Tiny ImageNet, and Fashion MNIST. For our experiments, we make use of various models and architectures for multiple comparisons, as shown in Table 1. For experiments with CIFAR-100 and TinyImagenet, we used ResNet18 as the student model when applying our knowledge distillation method, `ReffAKD`, with autoencoder while using ResNet50 as the teacher model for vanilla KD. For experiments with

Fashion MNIST, we opted to use LeNet-5 [13] as the student model due to the relatively lower complexity of the task at hand. In this case, we utilized ResNet18 as the teacher model for vanilla KD. The autoencoder architecture used for `ReffAKD` is described in Section 3.1.

Table 1: Compared teacher and student model of Vanilla KD for each dataset

|  | CIFAR-100 | Tiny ImageNet | Fashion MNIST |
|---|---|---|---|
| Teacher model | ResNet50 | ResNet50 | ResNet18 |
| Student model | ResNet18 | ResNet18 | LeNet5 |

For all the model architectures, we used Stochastic Gradient Descent with a momentum of 0.9, weight decay of 5e-4, an initial learning rate of 0.1 with step learning rate decay by 0.2, at intervals of 60, 120, and 160. All the models were trained for 200 epochs. All the experiments were performed on Nvidia A10 GPU with 24 GiB of memory. We also used standard data augmentation techniques with all the datasets - rotation by a maximum of 15 degree and random horizontal flipping. In the case of `ReffAKD` we used $\gamma$ of 0.0225, 0.0225, and 0.008 for CIFAR-100, Tiny Imagenet, and Fashion MNIST, respectively. For the knowledge distillation hyperparameters, temperature, $T$, and alpha ($\alpha$), we used grid search to find the best-performing combination. The search was conducted on $T = [1, 5, 10, 20]$ and $\alpha$ ranging from 0.1 to 0.9 with a step size of 0.1 as shown in Table 5. More discussion about these parameters is presented in Sec. 4.5.

## 4.2 Accuracy on CIFAR-100

Our experiments on CIFAR-100 show consistent improvement over the vanilla KD. For CIFAR-100, we achieve the best top 1 accuracy of $77.97\%$ with our approach, as compared to $77.57\%$ in the case of vanilla KD. The baseline ResNet18 accuracy is $76.08\%$ as shown in Table 2. This shows that using soft labels generated by `ReffAKD` improves the student model performance by a considerable amount from the baseline and generates similar to better performance as compared to using vanilla KD. In the case of CIFAR-100, we achieved the best performance for both `ReffAKD` and vanilla KD on $T = 5$ and $\alpha = 0.8$.

Table 2: Best accuracy on CIFAR-100 using ResNet18 (baseline), ResNet50 (baseline), Vanilla KD (ResNet50 to ResNet18), `ReffAKD`(ResNet18 student). Corresponding to Fig. 5 (a).

| ResNet18 (baseline) | ResNet50 (baseline) | Vanilla KD | ReffAKD |
|---|---|---|---|
| 76.08 | 77.31 | 77.57 | 77.97 |

In the case of CIFAR-100, we also outperform baseline ResNet50 accuracy, $77.31\%$, which demonstrates that knowledge distillation is an effective technique that can allow smaller networks to achieve better performance than larger models if tuned properly.

## 4.3 Accuracy on Tiny Imagenet

For Tiny Imagenet, we achieve the best top 1 accuracy of $63.67\%$ with our approach, as compared to $63.62\%$ in the case of vanilla KD. The baseline ResNet18 accuracy was $63.23\%$. Results on Tiny Imagenet also show improvement over ResNet18 baseline and vanilla KD accuracies. The amount of resources consumed also differs by a large amount which shows the efficacy of our approach on complex datasets like Tiny Imagenet. For our approach, i.e, using autoencoder-generated soft labels, we got the best accuracy with $T = 20$ and $\alpha = 0.9$, whereas, for vanilla KD, we obtained the best accuracy with $T = 10$ and $\alpha = 0.9$.

Table 3: Best accuracy on Tiny Imagenet using ResNet18 (baseline), ResNet50 (baseline), Vanilla KD (ResNet50 to ResNet18), `ReffAKD`(ResNet18 student). Corresponding to Fig. 5 (f).

| ResNet18 (baseline) | ResNet50 (baseline) | Vanilla KD | ReffAKD |
|---|---|---|---|
| 63.23 | 64.2 | 63.62 | 63.67 |

## 4.4    Accuracy on Fashion MNIST

For Fashion MNIST, we chose to use a smaller student model (LeNet-5) because of lower task complexity. LeNet-5, a convolutional network with two fully connected layers and two convolutional layers, obtains a top 1 accuracy of $92.14\%$ on the test set of Fashion MNIST when trained on hard labels. After using vanilla KD from ResNet18 with baseline accuracy of $95.3\%$, it achieves the best accuracy of $92.3\%$ ($T = 10$, $\alpha = 0.8$) while using `ReffAKD`, we are able to achieve top 1 accuracy of $92.37\%$ ($T = 20$, $\alpha = 0.6$). This shows that `ReffAKD` can also be efficiently used with smaller datasets while achieving similar performance. Our results on Fahion MNIST confirm that the results are consistent with the previous two datasets' cases.

Table 4: Best accuracy on Fashion MNIST using LeNet-5 (baseline), ResNet18 (baseline), Vanilla KD (ResNet18 to LeNet-5), `ReffAKD`(LeNet-5 student). Corresponding to Fig. 5 (k).

| LeNet5 (baseline) | ResNet18 (baseline) | Vanilla KD | ReffAKD |
|:---:|:---:|:---:|:---:|
| 92.14 | 95.3 | 92.3 | 92.37 |

## 4.5    Effects of Temperature and Alpha

The search for the best temperature and alpha involves systematically evaluating the performance of the knowledge distillation method using different combinations of temperature and alpha values. We have tabulated the accuracy results for different hyperparameter values for CIFAR-100 in Table 5, which shows the variation in accuracy of the two methods for different hyperparameter values. In the presented table, we have included alpha values ranging from 0.6 to 0.9 since the majority of performance gains occurred within this range. Conversely, we excluded alpha values ranging from 0.1 to 0.5 for both vanilla KD and `ReffAKD` as they did not exhibit significant improvements over the baseline model. After analyzing the results, we have observed that the highest test accuracy is achieved for a temperature value of 5 and an alpha value of 0.8. Furthermore, our experiments have revealed that knowledge distillation performs better with higher values of alpha, which is in line with previous research on knowledge distillation [7]. This suggests that using a high value of alpha can lead to more effective knowledge transfer from the teacher network to the student network. The experimental figures for the temperature and alpha for the other datasets are presented in the appendix.

Table 5: Accuracy on CIFAR-100 based on different values of temperature and alpha

| Temperature, $T$ | Alpha, $\alpha$ | ReffAKD accuracy | Vanilla KD accuracy |
|:---:|:---:|:---:|:---:|
| 20 | 0.6 | 76.80% | 76.84% |
|    | 0.7 | 76.89% | 77.06% |
|    | 0.8 | 77.49% | 77.20% |
|    | 0.9 | 76.75% | 76.98% |
| 10 | 0.6 | 77.27% | 77.03% |
|    | 0.7 | 76.85% | 77.34% |
|    | 0.8 | 77.50% | 77.09% |
|    | 0.9 | 76.52% | 77.42% |
| 5  | 0.6 | 77.22% | 77.08% |
|    | 0.7 | 76.84% | 77.54% |
|    | 0.8 | **77.97%** | **77.59%** |
|    | 0.9 | 76.64% | 77.30% |
| 1  | 0.6 | 75.94% | 76.58% |
|    | 0.7 | 76.17% | 77.01% |
|    | 0.8 | 76.53% | 76.86% |
|    | 0.9 | 75.92% | 76.85% |

## 4.6 Resource Consumption

We demonstrate the resource efficiency of our approach by showing a reduction in the number of parameters, FLOPs, MACs, and memory consumption against the teacher models (of vanilla KD) in Fig. 5. Our approach is able to drastically reduce resource consumption, yet achieve comparable or better performance as compared to vanilla KD which uses a larger model, while generating accurate soft labels utilizing information from the training set. Table 6 shows the reductions in resource utilization that `ReffAKD` is able to achieve. For CIFAR-100 and Tiny Imagenet, we obtain a similar

Table 6: Reduction in resource consumption in `ReffAKD`, corresponding to Fig. 5.

|  | FLOPs | MACs | Parameters | Memory |
|---|---|---|---|---|
| CIFAR100 | 354x | 358x | 501x | 503x |
| Tiny ImageNet | 354x | 355x | 530x | 533x |
| FashionMNIST | 154x | 155x | 239x | 237x |

amount of reduction in FLOPs and MACs because the same architecture is used - ResNet50 for vanilla KD, and CAE for `ReffAKD`. Also, note that although the input sizes for the two datasets are different (32x32 for CIFAR-100 and 64x64 for Tiny Imagenet), the "ratio," i.e., FLOPs in vanilla KD over FLOPs in ReffAKD, remains similar because of the same architectures. Note the Y-axes of the resource consumption in Fig. 5 are in a log scale.

## 4.7 Comparative analysis

In this section we present a comparative analysis of various KD techniques and ReffAKD. The aim is to provide an objective evaluation of our approach, while considering the resource efficiency, and compatability with existing KD methods.
To ensure a robust and fair comparision we fixed the KD and model training hyper-parameters accross all methods. The model hyper-parameters used are listed in Tab. 7. For the KD setup we

Table 7: Hyper-Parameters

| Batch Size | Epochs | LR Initial | LR Decay | Decay Stages | Momentum | Optimizer | Weight Decay |
|---|---|---|---|---|---|---|---|
| 256 | 200 | 0.1 | 0.2 | 60, 120, 160 | 0.9 | SGD | 0.0005 |

fixed temperature to $4$ and alpha to $0.9$. This standardized approach made it possible to identify the performance effect specific to KD process without emphasizing on hyper-parameter tuning.

**Resource efficiency and competitive performance** Tab. 8 shows that `ReffAKD` consistently achieves competitive performance across CIFAR-100 and Tiny Imagenet datasets. Importantly, this performance is achieved using significantly less resources as compared to the methods using larger teacher model as shown in Sec 4.6 and without tuning `ReffAKD`'s hyper-parameters. This makes `ReffAKD` significant in resource-constrained scenarios like embedded or edge devices.

Table 8: Accuracy Comparison on CIFAR-100 and Tiny Imagenet Datasets with ResNet 50 [6] , ResNet 18 [6] , AT [9] , KD [7] , FITNET [21] , KDSVD [14] , PKT [18] , RKD [17] and DKD [32]

|  | ResNet 50 | ResNet 18 | AT | KD | FITNET | KDSVD | PKT | RKD | DKD | DKD+ReffAKD | ReffAKD |
|---|---|---|---|---|---|---|---|---|---|---|---|
| CIFAR-100 | 76.28 | 75.42 | 75.16 | 76.91 | 73.08 | 75.56 | 76.47 | 76.28 | 77.49 | 77.14 | 77.03 |
| TinyImage | 63.68 | 62.66 | 62.57 | 63.40 | 62.02 | 63.54 | 61.20 | 62.51 | 64.32 | 64.75 | 63.57 |

**Compatability with exisiting techniques** `ReffAKD` exibits seamless compatability with any exisiting logit-based KD techniques. We demonstrate this in Tab. 8 by combining `ReffAKD` with DKD. We

see a notable performance gain using `ReffAKD`with DKD which could be further enhanced with hyper-parameter tuning. This compatibility suggests that researchers can capitalize on advancements in KD and implement `ReffAKD`alongside to get competitive performance in resource-constrained environments. While we deliberately refrained from hyper-parameter tuning in the case of integrating `ReffAKD`with DKD, our results hint at the potential for further improvement by tuning the temperature and alpha parameter.

## 5   Discussion and Future Work

The potential of autoencoders in finding efficient and effective one-dimensional image representations in computer vision is well-explored. Our approach capitalizes on this by utilizing a small and efficient autoencoder to generate a one-dimensional representation of the input and produce a soft probability distribution for knowledge distillation.

We believe that this methodology can also be readily applied to other domains, such as Natural Language Processing, where a small RNN-based autoencoder is utilized to derive an embedding of input sentences. This, in turn, could be employed to distill models such as TinyBERT [8] or other BERT [3] variants for classification by using `ReffAKD` to generate soft probability distribution. Our approach can also provide direct supervision to larger models, potentially leading to further performance gains, as our methodology does not rely on the existence of a large teacher model. Our experimentation exhibits that selecting the optimal hyperparameters, such as $\alpha$ and temperature, for vanilla KD or `ReffAKD` is not straightforward. While a high temperature, such as 5, 10, or 15, tends to work well within the range of alpha values from 0.6 to 0.9, this may vary across different datasets.

Also, for future investigation, the impact of different autoencoder architectures on distillation performance can be explored. In this paper, we utilized a specific autoencoder design that was optimized for small size and efficiency. However, it is possible that other architectures may yield better performance. For example, incorporating attention mechanisms into the design could potentially enhance the quality of the learned one-dimensional representations and improve distillation performance. Another important direction of future work is to evaluate the effectiveness of our approach with other existing approaches after appropriate hyper-parameter tuning, specifically alpha and temperature for KD. It would be interesting to see how these hyper-parameters behave not only within our approach but also in comparison to standard methods. Discovering unique trends in hyper-parameter settings can offer insights into the underlying dynamics of knowledge transfer. Understanding these interactions and their impact on our method's performance relative to others will enhance our approach and contribute to a broader understanding of hyperparameter tuning in model compression.

## 6   Conclusion

We introduce an innovative and resource-efficient method for generating soft labels to facilitate knowledge distillation in student models, specifically tailored for computer vision classification tasks. Our approach leverages a tiny autoencoder, minimizing resource consumption while achieving competitive or superior performance compared to traditional knowledge distillation methods that rely on larger teacher models. We conducted a comprehensive analysis of various resource consumption indicators, including FLOPs (Floating Point Operations), MACs (Multiply-Accumulate Operations), parameter counts, and memory usage, comparing our approach to standard knowledge distillation techniques. Across multiple experiments using three different datasets, our method, `ReffAKD`, consistently demonstrated significantly lower resource requirements while maintaining similar or even superior accuracy levels. Furthermore, we investigated the compatibility of `ReffAKD` with other logit-based knowledge distillation techniques. Our findings revealed that `ReffAKD` can deliver competitive performance while drastically reducing resource utilization. In summary, our research offers a valuable contribution to the field of deep learning by enabling researchers operating in resource-constrained environments to harness the benefits of knowledge distillation without the need for extensive resources to train large teacher models. This streamlines the knowledge distillation process and also opens doors for wider adoption and application of this effective technique.

## Acknowledgments and Disclosure of Funding

This publication is supported by the National Science Foundation under Grant No. 1945541 (transferred and extended to No. 2302610). Any opinions, findings, and conclusions or recommendations expressed in this material are those of the author(s) and do not necessarily reflect the views of the National Science Foundation.

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

# Appendix

**Effects of Temperature and Alpha on Tiny Imagenet and Fashion MNIST**

We experimented with similar temperature and alpha values on Tiny Imagenet, i.e $\alpha$ in range of $0.1$ to $0.9$ with step size of $0.1$ and $T$ of $20, 10, 5, 1$. As mentioned in 4.5 we only report $\alpha$ in range of $0.6$ to $0.9$ because the significant gains are shown only in these range.

In case of Tiny Imagenet as shown in Table 9 we achieve best accuracy of $63.67\%$ on $T = 20$, and $\alpha = 0.9$ for `ReffAKD`, whereas we receive best accuracy of $63.62\%$ on $T = 10$, and $\alpha = 0.9$ for vanilla knowledge distillation.

Table 9: Accuracy on Tiny Imagenet based on different values of temperature and alpha

| Temperature, $T$ | Alpha, $\alpha$ | `ReffAKD` accuracy | Vanilla KD accuracy |
|---|---|---|---|
| 20 | 0.6 | 63.42% | 63.39% |
|  | 0.7 | 62.96% | 63.31% |
|  | 0.8 | 62.96% | 62.96% |
|  | 0.9 | 63.67% | 63.29% |
| 10 | 0.6 | 62.95% | 63.41% |
|  | 0.7 | 62.76% | 63.04% |
|  | 0.8 | 63.12% | 63.59% |
|  | 0.9 | 63.37% | 63.62% |
| 5 | 0.6 | 63.28% | 62.64% |
|  | 0.7 | 62.45% | 63.12% |
|  | 0.8 | 63.01% | 63.22% |
|  | 0.9 | 63.08% | 63.18% |
| 1 | 0.6 | 62.77% | 63.14% |
|  | 0.7 | 62.41% | 62.83% |
|  | 0.8 | 63.13% | 63.15% |
|  | 0.9 | 63.07% | 62.99% |

In case of Fashion MNIST as shown in Table 10 we achieve best accuracy of $92.37\%$ on $T = 20$, and $\alpha = 0.6$ for `ReffAKD`, whereas we receive best accuracy of $92.30\%$ on $T = 10$, and $\alpha = 0.8$ for vanilla knowledge distillation.

Table 10: Accuracy on Fashion MNIST based on different values of temperature and alpha

| Temperature, $T$ | Alpha, $\alpha$ | `ReffAKD` accuracy | Vanilla KD accuracy |
|---|---|---|---|
| 20 | 0.6 | 92.37% | 92.00% |
|  | 0.7 | 92.13% | 91.97% |
|  | 0.8 | 92.23% | 92.06% |
|  | 0.9 | 92.31% | 92.01% |
| 10 | 0.6 | 92.02% | 91.95% |
|  | 0.7 | 92.08% | 92.18% |
|  | 0.8 | 91.93% | 92.30% |
|  | 0.9 | 92.06% | 92.08% |
| 5 | 0.6 | 92.11% | 92.08% |
|  | 0.7 | 92.13% | 91.83% |
|  | 0.8 | 92.24% | 92.20% |
|  | 0.9 | 92.02% | 92.15% |
| 1 | 0.6 | 91.96% | 92.04% |
|  | 0.7 | 91.74% | 91.64% |
|  | 0.8 | 91.31% | 91.39% |
|  | 0.9 | 89.78% | 91.60% |

