# OpenReview forum: "ReffAKD: Resource-efficient Autoencoder-based Knowledge Distillation"
_NeurIPS.cc/2023/Workshop/WANT — WANT@NeurIPS 2023 Poster_

### Official Review · Reviewer_fswH · 2023-10-25

**Confidence:** 4

**Review:**

**Summary**

The paper proposes an approach for knowledge distillation that eliminates the need for a large teacher model. The authors employ a compact autoencoder to extract essential features and calculate similarity scores between different classes. The proposed method is evaluated on several benchmark datasets, and the results show that it achieves similar or even superior performance compared to standard knowledge distillation methods.

**Soundness** 1

**Presentation** 3

**Contribution** 1

**Strengths**

The authors provide a detailed description of the approach. The area is important.

**Weakness**

All the numbers provided in Figures 5 and Tables 2 and 3 are based on one random seed. Given the small difference, I would require the authors to provide mean and standard deviation results under multiple seeds.

The authors claim that "we explore an efficient, unsupervised method to identify class similarities and generate high-quality soft labels for KD." However, it still requires randomly selecting 40 samples from each of the different classes to calculate the soft label. The method does not seem unsupervised to me.

**Questions**

Could the authors give an explanation of how they add the skip connection to the three-layer autoencoder?

Why do the authors claim that "our approach falls within the realm of knowledge distillation rather than being a generic soft label generator"?

**Limitations** Not discussed.

**Flag For Ethics Review**: No ethics review needed.

**Rating**: 3: Reject

---

### Official Review · Reviewer_dDuB · 2023-10-26
**Writing of the paper needs improvement and results on larger datasets are required to strengthen the claims.**

**Confidence:** 4

**Review:**

Summary:

The paper proposes an auto-encoder based technique to efficiently generate soft-labels for knowledge distillation and eliminate the need for a large teacher model. Experimental results were shown on small datasets such as CIFAR-100, Fashion-MNIST and Tiny-Imagenet.

Strengths:
1. Using auto-encoders as a proxy for knowledge distillation is an interesting idea. The paper also compares the proposed method to multiple existing KD approaches.
2. Extensive ablations on the impact of alpha and the temperature parameter on final accuracy provides some good insights.

Weaknesses:
1. Writing needs to be improved significantly and the paper has several typos formatting issues.
2. Results in Figure 5 are not clear. The figure has references to ‘TinySLG’, but not referenced anywhere else in the paper.
3. Additionally, it is important to compare and highlight the wall-clock training time needed for a large teacher model and the proposed auto-encoder, across different datasets. This is currently missing in the paper.
4. Results were only shown on smaller datasets and it is unclear how this approach will scale to larger datasets and tasks. It would also help to include other Transformer based networks like ViT to strengthen the claims.

---

### Meta-Review · Area_Chair_g6sy · 2023-10-27

**Recommendation:** Accept (Poster)
**Confidence:** 4

**Metareview:**

The papers received mixed reviews, with Reviewer fswH recommending for rejection. However, no strong arguments are made for rejection beyond requiring re-running the experiments with multiple seeds and discussion regarding wherever the method is unsupervised while not.
As the idea itself is somewhat interesting, despite some notable shortcomings, I am inclined to recommend the paper for acceptance and foster discussion around this topic.

That being said, the authors should closely follow the comments made by the reviewers, adding results for multiple seeds, improve the overall quality of the presentation and if possible, run experiments on larger datasets too.

---

### Decision · Program_Chairs · 2023-10-28

**Decision:**

Accept (Poster)

**Comment:**

We thank the authors for their time and contribution to WANT and we are pleased to share that after the reviewing process the paper has been accepted. Congratulations! We encourage the authors to consider reviewers' feedback for the improvement of the camera-ready version. We hope to see you in person at the workshop and brainstorm on efficient training research together!